# *Lindera obtusiloba* Attenuates Oxidative Stress and Airway Inflammation in a Murine Model of Ovalbumin-Challenged Asthma

**DOI:** 10.3390/antiox9070563

**Published:** 2020-06-27

**Authors:** Ba-Wool Lee, Ji-Hye Ha, Han-Gyo Shin, Seong-Hun Jeong, Ju-Hong Kim, Jihye Lee, Ji-Young Park, Hyung-Jun Kwon, Kyungsook Jung, Woo-Song Lee, Young-Bae Ryu, Jae-Ho Jeong, In-Chul Lee

**Affiliations:** 1Functional Biomaterial Research Center, Korea Research Institute of Bioscience and Biotechnology, Jeongeup-si, Jeollabuk-do 56212, Korea; pl0706@kribb.re.kr (B.-W.L.); jihye2640@kribb.re.kr (J.-H.H.); shangyo48@naver.com (H.-G.S.); jsh0830@kribb.re.kr (S.-H.J.); wnghd93@kribb.re.kr (J.-H.K.); et1118@kribb.re.kr (J.L.); loveme@kribb.re.kr (J.-Y.P.); hjkwon@kribb.re.kr (H.-J.K.); jungks@kribb.re.kr (K.J.); wslee@kribb.re.kr (W.-S.L.); ybryu@kribb.re.kr (Y.-B.R.); 2Department of Veterinary Pharmacology and Toxicology, College of Veterinary Medicine, Chonnam National University, Gwangju 61186, Korea; 3Department of Microbiology, Chonnam National University Medical School, Gwangju 61186, Korea

**Keywords:** *Lindera obtusiloba*, allergic asthma, anti-oxidants activity, nuclear factor-erythroid 2-related factor, mitogen-activated protein kinases, nuclear factor-kappaB

## Abstract

*Lindera obtusiloba* is widespread in northeast Asia and used for treatment of improvement of blood circulation and anti-inflammation. In this study, we investigated anti-inflammatory and anti-oxidant effects of the methanolic extract of *L. obtusiloba* leaves (LOL) in an ovalbumin (OVA)-challenged allergic asthma model and tumor necrosis factor (TNF)-α-stimulated NCI-H292 cell. Female BALB/c mice were sensitized with OVA by intraperitoneal injection on days 0 and 14, and airway-challenged with OVA from days 21 to 23. Mice were administered 50 and 100 mg/kg of LOL by oral gavage 1 h before the challenge. LOL treatment effectively decreased airway hyper-responsiveness and inhibited inflammatory cell recruitment, Th2 cytokines, mucin 5AC (MUC5AC) in bronchoalveolar lavage fluid in OVA-challenged mice, which were accompanied by marked suppression of airway inflammation and mucus production in the lung tissue. LOL pretreatment inhibited the phosphorylation of mitogen-activated protein kinases (MAPKs) and nuclear factor-kappa B (NF-κB) with suppression of activator protein (AP)-1 and MUC5AC in the lung tissue. LOL also down-regulated expression of inflammatory cytokines, and inhibited the activation of NF-κB in TNF-α-stimulated NCI-H292 cells. LOL elevated the translocation of nuclear factor-erythroid 2-related factor (Nrf-2) into nucleus concurrent with increase of heme oxyngenase-1 (HO-1) and NAD(P)H quinine oxidoreductase 1 (NQO1). Moreover, LOL treatment exhibited a marked increase in the anti-oxidant enzymes activities, whereas effectively suppressed the production of reactive oxygen species and nitric oxide, as well as lipid peroxidation in lung tissue of OVA-challenged mice and TNF-α-stimulated NCI-H292 cells. These findings suggest that LOL might serve as a therapeutic agent for the treatment of allergic asthma.

## 1. Introduction

Allergic asthma is a chronic inflammatory disease that has substantially increased in prevalence over the recent decades [1]. In general, allergic asthma is caused by the inhalation of allergens, including pollen, house dust, inhalants, and air pollutants. Additionally, it is characterized by eosinophil-rich airway inflammation, mucus hypersecretion with goblet cell hyperplasia, and airway hyper-responsiveness [2]. Exposure to different allergens causes an imbalance between T helper (Th) 1 and Th2 responses, resulting in asthmatic responses in the airway [3,4]. Inflammatory cells, particularly eosinophils, produce cytokines, chemokines, and growth factors that cause persistent inflammatory responses and mucus hypersecretion in the airway [5]. Of these factors, Th2 cells contribute to the pathogenesis of asthma by stimulating B cells that produce the allergen-specific immunoglobulin E (IgE), and by recruiting the eosinophils and other inflammatory cells into the airways [5,6]. In addition, inflammatory cells participate in the release of Th2 cytokines, such as interleukin (IL)-4, IL-5, and IL-13, which play a crucial role in the development and progression of allergic asthma [7].

Oxidative stress is a status of imbalance between pro-oxidants and anti-oxidants, which is the outcome of the overproduction of reactive oxygen species (ROS). It has been reported that ROS-mediated oxidative stress contributes to the pathogenesis of allergic asthma [2,8]. The nuclear factor-erythroid 2-related factor (Nrf-2) is a redox-sensitive transcription factor that protects oxidative stress through the activation of anti-oxidant response element (ARE) genes, resulting in up-regulation of anti-oxidant enzymes, such as heme oxygenase-1 (HO-1) and NAD(P)H quinine oxidoreductase 1 (NQO1) [9,10]. HO-1 is an enzyme that catalyzes the oxidative degradation of heme into bilirubin and free iron [2,11]. NQO1 is a reductase, which induces reduction reaction from quinones to hydroquinones, regulating ROS generation [11]. The activating of Nrf-2/HO-1/NQO1 confer a protection to tissues and cells from oxidative stress-related damage. Previous studies indicate that regulation of ROS production through Nrf-2 signaling have a protective role in the ovalbumin (OVA)-challenged asthma model [2,12].

The ROS-mediated activation of nuclear factor-kappa B (NF-κB) and mitogen-activated protein kinases (MAPKs) induce important roles of development and progression of asthma [13,14]. The NF-κB is a multicellular transcription factor, which plays inflammatory responses by various inflammatory mediators, cytokines, and mucin 5AC (MUC5AC) in asthmatic responses [15,16,17]. On the other hand, MAPKs are composed of three protein kinases, including extracellular-signal-regulated kinase (Erk), c-Jun N-terminal kinase (JNK), and p38MAPK. MAPKs play a critical role in the activation of inflammatory responses via intra- or extracellular signals, and link to the activation of activator protein (AP)-1 [18,19,20]. The AP-1 transcription factor complex is one of the major regulators of both proliferative and inflammatory responses [21]. The activation of AP-1 elevates MUC5AC expression and Th2 cytokines production [22,23,24]. In asthmatic conditions, up-regulation of Th2 cytokines and MUC5AC is associated with eosinophil recruitment, goblet cell hyperplasia, and airway limitation [5,6,25]. Therefore, the inhibition of NF-κB and MAPKs/AP-1 may be a synergistic treatment strategy for controlling asthmatic responses. Asthmatic patients have been treated with immunosuppressants, bronchodilators, and antihistamines. However, the use of these drugs is limited by its adverse effects. Recently, increased attention has been paid toward the exploration of plant-derived medicines for treating respiratory diseases, including airway inflammation, asthma, and chronic obstructive pulmonary disease [26,27,28].

Oxidative stress plays a crucial role in pathological conditions, including inflammation, cancer, metabolic and immune disorders [29,30]. Many researchers have reported that plant extracts are prone to modulate cellular oxidative status in several pathological conditions as mentioned above [30,31,32]. The bioactive phytochemicals or secondary metabolites from plant extracts possess various biological activities, such as anti-oxidant, anti-inflammatory, and antitumor both in vitro and in vivo [31,32]. Recently, with the current upsurge of interest in efficacy of plant extracts, active substance extracted and concentrated from plants (as single or combined phytocomplexes), such as polyphenols, flavonoids, and alkaloids, have been extensively investigated to proven their clinical efficacy for pharmaceutical application [29,32]. Furthermore, it can be used to prevent or cure pathological conditions when they have proven safety and beneficial properties, as well as better bioavailability [29]. *Lindera obtusiloba*, a member of the Lauraceae family, is widespread in northeast Asia. *L. obtusiloba* has been used as herbal medicine for improving blood circulation and treating inflammation [33]. Previous studies demonstrated that the extract of *L. obtusiloba* has potent anti-oxidative and anti-inflammatory activities on atopic dermatitis and anti-allergic responses in mast cells [33,34,35,36]. In detail, *L. obtusiloba* has effects such as anti-inflammation and anti-allergy through the suppression of histamines and Th2 cytokines in a DFE/DNCB-induced atopic dermatitis model [34]. On the other hand, *L. obtusiloba* inhibited ROS generation in the Fe^3+^-EDTA/H_2_O_2_ system, and has potent 2,2-diphenyl-1-picryl hydrazyl (DPPH) radical scavenging activity [35]. Although many studies have reported the anti-oxidant and anti-inflammatory properties of *L. obtusiloba*, its protective effects of allergic asthma have not been elucidated. Thus, we investigated the effects of the methanol extract of LOL on OVA-challenged asthma murine model and TNF-α-stimulated NCI-H292 cells.

## 2. Materials and Methods

### 2.1. Ultra-Performance Liquid Chromatography-Quadrupole-Time of Flight Mass Spectrometry (UPLC Q-ToF/MS) Analysis

*Lindera obtusiloba* was collected from Jeju-do (Jeju, Republic of Korea) and purchased from The Korea Plant Extract Bank of Korea Research Institute of Bioscience and Biotechnology (KRIBB, PB2899.2). The tentative identification of compounds in methanolic extract of LOL was analyzed by ACQUITY UPLC system coupled with Vion IMS Q-ToF mass spectrometer (Waters Corp., Milford, MA, USA) using BEH C18 column (2.1 × 100 mm, 1.7 μm). The sample was injected (2 μL) and the elution was completed in 25 min with an isocratic flow rates of 0.4 mL/min at 35 °C. The mobile phases consisted of solvent A (0.1% formic acid in water, *v*/*v*) and solvent B (100% acetonitrile). The program began with isocratic elution with 95% solvent A (0–1 min), and then linear gradient was used for 20 min, lowering solvent A to 0%; from 20 to 22.30 min, the gradient returned to the initial composition (95% A), and then is was held constant to re-equilibrate the column. The analysis was carried out using negative mode, data-dependent mass spectrometry (MS) scanning from 100 to 1500 Da with a 0.2 s scan. The optimized MS conditions were as follow: capillary voltage of 2300 V, cone voltage of 40 V, desolvation temperature of 350 °C, desolvation gas flow of 800 L/h (N_2_) and source temperature of 110 °C. Leucine-enkephalin was used as the lock mass ([M−H]^−^
*m*/*z* 554.2615). The full scan data and MS/MS spectra were acquired using a UNIFI scientific information system (Traditional medicine library, Waters Corporation, Milford, MA, USA).

### 2.2. Test Compound and Reagents

The human lung epithelial cell line, NCI-H292 cell, was purchased from American type culture collection (ATCC; Rockyville, MD, USA). Dexamethasone (DEX) and OVA were purchased from Sigma-Aldrich (St. Louis, MO, USA) and human recombinant tumor necrosis factor alpha (TNF-α) was obtained from Peprotech (Rocky Hill, NJ, USA). The enzyme-linked immunosorbent assay (ELISA) kits for TNF-α, IL-4, IL-5, IL-6, IL-13, and eotaxin (R&D system, Minneapolis, MN, USA), MUC5AC (Cusabio Biotech Co. Wuhan, China) and IgE (BioLegend, CA, USA) were used according to the manufacturer’s instructions. The Diff-Quik^®^ stain kit and Periodic acid-Schiff (PAS) kit was purchased from IMEB Inc. (San Marcos, CA, USA).

### 2.3. Experimental Procedure

Specific pathogen-free female BALB/c mice (7 weeks old, 20–25 g) were obtained from Orient Bio (Seongnam, Republic of Korea). Previous studies demonstrated that female mice are more sensitive to development of allergic inflammation in airway than male mice [37]. Thus, we used female BALB/c mice to develop the OVA-challenged asthma model based on previous studies [2,37]. Mice were housed in groups of three or four under standard conditions with temperature 22 ± 2 °C, humidity 55 ± 5%, and 12 h light/dark cycle. Commercial rodent chow and water was supplied to mice *ad libitum*. All procedures of experiment were approved by the Institutional Animal Care and Use Committee of the Korea Research Institute of Bioscience and Biotechnology (Approval number: KRIBB-AEC-18234). Female BALB/c mice were randomly divided into the following 5 groups (*n* = 7 per group).
Normal control (NC) group: treated with vehicle (2% DMSO) from day 21 to day 25 and PBS sensitization/challengeOVA group: treated with vehicle (2% DMSO) from day 21 to day 25 and OVA sensitization/challengeDEX group: treated with 3 mg/kg from day 21 to day 25 and OVA sensitization/challengeLOL50 group: treated with LOL 50 mg/kg and OVA sensitization/challengeLOL100 group: treated with LOL 100 mg/kg and OVA sensitization/challenge

Mice were sensitized on days 0 and 14 via an intraperitoneal injection of 20 μg of OVA emulsified in 2 mg of aluminum hydroxide gel in 200 μL of phosphate-buffered saline (PBS). At days 21–23 after initial sensitization, mice were administered the treatments once daily by oral gavage. At the time of oral administration, mice were challenged for 1 h with OVA (1%, w/v, in PBS) using an ultrasonic nebulizer (NE-U12; OMRON Corp., Tokyo, Japan). DEX, as a positive control, was administered to mice at a dose of 3 mg/kg body weight [38].

### 2.4. Measurement of Airway Hyper-Responsiveness

To analyze airway hyper-responsiveness, mice were anesthetized, tracheostomized, and ventilated with Flexivent (SCIREQ Scientific Respiratory Equipment Inc., Montreal, PQ, Canada) [39] 24 h after the last OVA challenge. After baseline measurements of impedance (Zrs), methacholine (5, 10 and 20 mg/mL) or PBS was delivered (Aeroneb; SCIREQ) for 10 s. Airway hyper-responsiveness was measured every 30 s for one min and refreshed for 2 min. The volume history of the lung was established with 6 s deep inflations to a pressure limit of 30 cmH_2_O and then snapshot perturbation maneuver was imposed to measure the levels of airway hyper-responsiveness.

### 2.5. Bronchoalveolar Lavage Fluid (BALF) Collection and Inflammatory Cell Count

BALF collection was obtained as previously described [40]. Briefly, to obtain BALF, ice-cold PBS (0.7 mL) was infused was into lung two times and withdrawn each time through cannulation (total volume 1.4 mL). The BALF was centrifuged at 1000 rpm for 10 min at 4 °C; the supernatant was collected and used for ELISA. Whole cell pellet from the BALF was attached on a slide using Cytospin 4 centrifuge at 1000 rpm for 5 min (Thermo Scientific, Waltham, MA, USA). The BALF was stained with Diff-Quik^®^ staining reagent for differential cell count. Each slide was captured with a light microscope (Leica DM5000B, Wetzlar, Germany) under 40× objective lens. The number of total cells, eosinophils, macrophages, and other cells (neutrophils and lymphocyte) were presented as mean ± standard deviation (SD).

### 2.6. Measurement of Th2 Cytokines, Eotaxin in BALF, and OVA-Specific IgE in Serum

The levels of IL-4, IL-5, IL-13, eotaxin (R&D system), and MUC5AC (Cusabio Biotech Co., Wuhan, China) in BALF were measured using ELISA kits, according to the manufacturer’s instructions. The absorbance of sample was measured at 450 nm using a microplate reader (Bio-Rad Laboratories, Richmond, CA, USA).

After BALF collection, the blood samples were collected from the inferior vena cava and centrifuged at 3000 rpm for 10 min at 20 °C. The concentrations of total IgE and OVA-specific IgE in serum were measured using ELISA kits (BioLegend) according to the manufacturer’s instructions. The microtiter plates were coated with anti-IgE antibodies (anti-mouse IgE; 10 g/mL; BioLegend) in tris-buffered saline containing 0.05% Tween 20 (TBST) and incubated with a serum sample. After the plates were washed 4 times, and 200 μL o-phenylenediamine dihydrochloride (Sigma-Aldrich, St. Louis, MO, USA) was added to each well. The plates were incubated for 10 min in the dark conditions, and the absorbance was measured at 450 nm using a microplate reader (iMark^TM^, Bio-Rad Laboratories, Richmond, CA, USA).

### 2.7. Histopathological Examination of Lungs

The lung tissues were fixed in 10% (*v*/*v*) neutral-buffered formalin. To estimate the amount of airway inflammation or mucus production, the lung tissues were embedded in paraffin, cut into 4 μm thick sections, and stained with hematoxylin and eosin (H&E) (BBC Biochemical, Mount Vemon, WA, USA) or PAS (IMEB Inc., San Marcos, CA, USA), respectively. Quantitative analysis of airway inflammation and mucus production was performed in a blinded manner with a light microscope (Leica Microsystem) at the 10× and 20× objective lens. The degree inflammation and mucus production of each slide were graded (0, no lesions; 1, minimal; 2, mild; 3, moderate; and, 4, severe) and the index levels of inflammation and mucus production were represented as mean ± SD.

### 2.8. Immunoblotting in Lung Tissues

The lung tissues were homogenized (1/10, *w*/*v*) using a homogenizer in CER I (NE-PER Nuclear and Cytoplasmic Extraction Reagent, Thermo Scientific, Waltham, MA, USA). The supernatants were mixed with ice-cold CER II and incubated at 1 min. The samples were vortexed and centrifuged at 16,000× *g* for 10 min. The supernatants were collected (cytoplasmic extract) and mixed with NER and vortexed for 15 s every 10 min, for a total of 40 min. The samples were centrifuged at 16,000× *g* for 10 min and collected supernatants (nuclear fraction). The cytoplasmic and nuclear fractions were determined using a protein assay reagent (Bradford reagent; Bio-Rad Laboratories). Equal quantities of protein (30 μg) were separated by SDS-polyacrylamide gel (4–12%) electrophoresis and transferred to polyvinyl difluoride membranes. Membranes were incubated with a blocking solution (Thermo Scientific), and incubated with primary antibodies at 4 °C overnight. The membranes were washed three times with tris-buffered saline containing Tween 20 (TBST), and incubated with a horseradish peroxidase (HRP)-conjugated secondary antibody (1:10,000 dilution; Cell Signaling Technology) at room temperature for 1 h. The membranes were again washed with TBST, and then developed using an enhanced chemiluminescence kit (Thermo Scientific). For quantitative analysis, densitometric band values were determined using chemiluminescent scanner (LI-COR Biosciences, Lincoln, NE, USA). The primary antibodies and dilutions: p65NF-κB, p-p65NF-κB, JNK, p-JNK, Erk, p-Erk, p38MAPK, p-p38MAPK, c-Jun, c-Fos, β-actin (1:1000 dilution; Cell Signaling, MA, USA), MUC5AC, Nrf-2, HO-1, and NQO1 (1:1000 dilution; Abcam, Cambridge, UK).

### 2.9. Measurement of Nitric Oxide (NO) and ROS Level

The lung tissues were grinded using a homogenizer in ice-cold PBS (1/10, *w*/*v*). The homogenized lung tissues were centrifuged at 11,000× *g* for 15 min at 4 °C and collected to supernatants. The nitric oxide was determined using a nitric oxide plus kit (iNtRoN Biotechnology, Seoul, Republic of Korea). The absorbance was measured at 550 nm using microplate reader (Bio-Rad Laboratories). The ROS production was monitored using CellRox^®^ green reagent (Life technologies, Rockville, NY, USA), which could be used to evaluate ROS production. The samples were incubated with CellRox^®^ green reagent (5 μM) for 30 min at 37 °C. After incubated, ROS level was measured by measuring the fluorescence at 485 nm excitation and 520 nm emission on a fluorescence plate reader (Perkin–Elmer, Foster city, CA, USA).

### 2.10. Oxidative Stress Markers Analysis

The supernatants of grinded lung tissue were used for the measurement to glutathione (GSH) using (EZ-GSH assay kit, Dogen, Republic of Korea), thiobarbituric acid-reactive substances (TBARS, EZ-TBARS assay kit, Dogen, Republic of Korea), superoxide dismutase (SOD; EZ-SOD assay kit, Dogen, Republic of Korea) and catalase activity (CAT; EZ-Catalase assay kit, Dogen, Republic of Korea).

The lipid peroxidation was evaluated by TBARS assay, which measured the number of samples reacting with trichloroacetic acid. The supernatants were added with trichloroacetic acid and centrifuged at 3000× *g* for 10 min at 4 °C. The samples were added with indicator solution and incubated at 65 °C for 45 min. The absorbance was determined at 540 nm using microplate readers and values were expressed in nmol/mg protein.

Determination of the GSH concentration in the grinded lung samples were based on the reaction of GSH with 5,5′-dithiobis-2-nitrobenzoid acid (DTNB; chromogen) and GSH-reductase. Nicotinamide adenine dinucleotide phosphate (NADPH) solution was added in each sample and incubated 3 min at room temperature. The absorbance was measured using microplate readers at 415 nm. The GSH concentration was calculated by using the GSH assay kit according to the manufacturer’s instructions, and expressed nmol/mg protein.

The SOD activity in in the lung tissues were determined using the EZ-SOD kit, lung samples (20 μL) were mixed with water-soluble tetrazolium salt (WST) working solution (200 μL) and incubated with enzyme working solution at 37 °C for 20 min. The absorbance was determined as 450 nm. The SOD activity was normalized to the value of the control.

The CAT activity was measured by quantifying the fluorescence at 540 nm (excitation)/590 nm (emission). The samples were added to hydrogen peroxide (40 μM) in reaction buffer (25 μL) and incubated for 30 min. After incubation, each sample was incubated with oxi-probe/HRP working solution at 37 °C for 30 min. The CAT activity was normalized to the value of the control.

### 2.11. Cell Culture and Cell Viability Assay

NCI-H292 cells were incubated in Roswell Park Memorial Institute (RPMI) 1640 media (Gibco, San Diego, CA, USA) with 10% heat-inactivated fetal bovine serum (FBS; Gibco) and 1% antibiotics at 37°C in a 5% CO_2_ incubator. The cells were seeded into 96-well plates at a density of 5 × 10^4^ cells/well and incubated in RPMI1640 (0.1% FBS) in the presence of different concentrations of LOL (0, 25, 50, and 100 μg/mL). After incubation for 24 h, the effect of LOL on cell viability was measured by the water-soluble tetrazolium salt 1 assay solution (WST-1; EZ-CyTox, Dogen, Republic of Korea). The WST-1 solution was added to each well (10% of total volume) and incubated for 1 h, absorbance was determined using a microplate reader (Bio-Rad Laboratories) at 450 nm.

### 2.12. Measurement of 2,2-Diphenyl-1-Picryl Hydrazyl (DPPH) Radical Scavenging Activity

DPPH (Sigma-Aldrich) radical scavenging assay was performed as described previously with Abderrahim et al. [41]. Different concentrations of LOL (0, 12.5, 25, 50, and 100 μg/mL, 100 μL) were mixed with 200 μM DPPH radical solution (dissolved in methanol, 100 μL) in 96-well microplate for 30 min at room temperature. The absorbance was measure at 520 nm with a blank containing DPPH and methanol. DPPH radical scavenging activity (%) was calculated by using following equation: (1 − (*As*/*Ac*)) × 100, where *As* is the absorbance in the present of sample and *Ac* is the absorbance in the absence of sample.

### 2.13. Measurement of Levels of Pro-inflammatory Cytokine Production and ROS, and Oxidative Stress Marker in TNF-α-Stimulated NCI-H292 Cells

To further therapeutic application of LOL for asthma treatment, TNF-α-stimulated human lung epithelial cells, NCI-H292, has been used to evaluate the anti-inflammatory effects in vitro corresponding to OVA-challenged asthma mice model [42,43,44]. The cells (5 × 10^4^ cells/well) were seeded in 6-well plates in RPMI media, treated with LOL (0, 25, 50, and 100 μg/mL) and DEX (20 μg/mL) for 1 h, and then incubated in the presence of human recombinant TNF-α 30 ng/mL for 24 h. The concentrations of IL-6 and TNF-α (24 h) in the culture medium were quantified using a competitive ELISA kit (R&D system) according to the manufacturer’s instructions. The absorbance was measured at 450 nm in a microplate reader (Bio-Rad Laboratories). The cells were collected and used to analyze ROS production and oxidative stress makers, such as GSH and SOD, according to manufacturer’s instruction as described above.

### 2.14. Immunoblotting in TNF-α-Stimulated NCI-H292 Cells

The cells were treated with LOL as described above and then in the presence of TNF-α (30 ng/mL) for 1 h. The cells collected to cytoplasmic and nuclear proteins by using NE-PER Nuclear and Cytoplasmic Extraction Reagents (Thermo Scientific). The protein concentration for each sample was determined using a protein assay reagent (Bradford reagent; Bio-Rad Laboratories). The levels of p-p65NF-κB, Nrf-2, HO-1 and β-actin (loading control) were determined using immunoblotting.

### 2.15. Quantitative Real-Time Polymerase Chain Reaction (PCR) Analysis of Cytokines

The cells were incubated with LOL (25, 50 and 100 μg/mL) and DEX (20 μg/mL) followed by TNF-α (18 h). Thereafter, the cells were washed with PBS and total ribonucleic acid (RNA) was extracted using an RNA extraction kit (Qiagen, Valencia, CA, USA). The RNA concentration and purity were measured microphotometer (Allsheng Instruments Co., Hangzhou, China). For real-time PCR, total RNA (1 μg) was used for cDNA synthesis, employing iScript cDNA synthesis kit (Bio-Rad Laboratories). Real-time PCR was performed in triplicate with the CFX96 Touch^TM^ Real-time PCR detection system (Bio-Rad Laboratories). A SensiFast^TM^ SYBR No-ROX kit (BioLine, Tauton, MA, USA) was used to prepare the substrate for PCR. The following primer sequences were used: IL-4 forward; 5′- ATG GGT CTC ACC TCC CAA CT -3′, IL-4 reverse; 5′- TAT CGC ACT TGT GTC CGT GG-3′ (Gene-Bank accession number: NM_172348.3), IL-5 forward; 5′- CAG GGA ATA GGC ACA CTG GA -3′, IL-5 reverse; 5′- TCT CCG TCT TTC TCC ACA C -3′ (Gene-Bank accession number: NM_000879.3), IL-13 forward; 5′- TGG TAT GGA GCA TCA ACC TGA C -3′, IL-13 reverse; 5′- GCA TCC TCT GGG TCT CG -3′ (Gene-Bank accession number: NM_001354993.2), GAPDH forward; 5′-ATC ACC ATC TTC CAG GAG CGA-3′, GAPDH reverse; 5′-AGG GGC CAT CCA CAG TCT T-3′ (Gene-Bank accession number: NM_001357943.2) according to the manufacturer’s instructions.

### 2.16. Statistical Analysis

Data were presented as the means ± SD and analyzed using GraphPad Prism 5 (GraphPad Software, CA, USA). Statistical analysis was performed using analysis of variance by a Tukey’s multiple comparison test. *p* ≤ 0.05 were considered be have statistical significance.

## 3. Results

### 3.1. Tentative Characterization of LOL Extract

LOL were extracted using 100% methanol and analyzed by UPLC Q-ToF/MS (Figure 1 and Table 1). The compounds of LOL were tentatively identified according to information derived from MS deprotonated molecules ([M−H]^−^) and the fragmentation pattern of mass spectra compared with previous literature in negative mode. In previous studies, the kaempferol and quercetin are representative derivatives in many natural products [45,46,47]. Peak 1 exhibited [M−H]^−^ (447 Da) and [M−146−H]^−^ (301 Da), which were characteristic fragment of quercetin at *m/z* 301 and rhamnoside (146 Da) in negative mode [46,47]. The fragment patterns of peak 2 demonstrated [M−H]^−^ at *m*/*z* 431 and [M−146−H]^−^ at *m*/*z* 282. Kaempferol and rhamnoside have molecular weights of 285 and 146, respectively, and peak 2 was tentatively identified as kaempferol rhamnoside [47,48]. The contents of quercetin rhamnoside and kaempferol rhamnoside determined to be 26.04 ± 0.05 mg/g (2.60%) and 15.82 ± 0.02 mg/g (1.58%), respectively (Table 1).

### 3.2. Effects of LOL on Airway Hyper-Responsiveness in OVA-Challenged Asthma Model

The airway hyper-responsiveness of OVA-challenged mice was higher than that of NCs at any concentrations, especially at 10 and 20 mg/mL of methacholine (Figure 2). The DEX-treated mice had showed lower airway hyper-responsiveness than the OVA-challenged mice. In addition, the LOL-treated groups exhibited a marked reduction airway hyper-responsiveness in a dose-dependent manner compared to the OVA-challenged group, especially at 20 mg/mL methacholine.

### 3.3. Effects of LOL on Inflammatory Cell Counts, Th2 Cytokines, Eotaxin, MUC5AC of BALF, and OVA-Specific IgE of Serum in OVA-Challenged Asthma Model

As shown in Figure 3, the OVA-challenged mice had a significant increase in the number of eosinophils, macrophages, and other inflammatory cells compared to the NCs (Figure 3A). The levels of IL-4, IL-5, IL-13, eotaxin, and MUC5AC significantly increased in BALF collected from OVA-challenged mice compared to the NCs (Figure 3B). OVA-challenged mice had a significant increase in total IgE and OVA-specific IgE in serum (Figure 3C). In contrast, LOL-treated mice exhibited a significant decrease in inflammatory cells, in particular eosinophils and macrophages, in the BALF compared with the OVA-challenged mice. LOL-treated mice had a significant decrease in the levels of IL-4, IL-5, IL-13, eotaxin, and MUC5AC in the BALF. In addition, LOL treatment showed a significant decrease in the total IgE and OVA-specific IgE levels in the serum compared with those of OVA-challenged mice in a dose-dependent manner.

### 3.4. Effects of LOL on Inflammatory Response and Mucus Production in the Lung Tissue from the OVA-Challenged Asthma Model

OVA-challenged asthmatic lung showed a marked infiltration of inflammatory cells into the peribronchiolar and perivascular lesions compared to that of the lung tissue in the NC (Figure 4A). Moreover, lung tissues from OVA-challenged mice stained with PAS showed overproduction of mucus with goblet cell hyperplasia (Figure 4B). In contrast, LOL-treated mice exhibited less infiltration of inflammatory cells into the peribronchiolar and perivascular lesions concurrent with reduction of inflammation index levels compared to the OVA-challenged mice. The LOL-treated mice also exhibited a reduction of in mucus production and goblet cell hyperplasia corresponding to a significant decrease in the levels of mucus production index.

### 3.5. Effects of LOL on MAPKs/AP-1, p65NF-κB and MUC5AC in the Lung Tissue of the OVA-Challenged Asthma Model

To further examine the underlying mechanism of LOL, we determined the effects of LOL on the MAPKs/AP-1, p65NF-κB and MUC5AC expression in the lung tissues (Figure 5). As presented in Figure 5, OVA-challenged mice had a remarkable increase in the phosphorylation of MAPKs (JNK, Erk, p38MAPK), with elevation of AP-1 (c-Jun and c-Fos) compared with normal control mice (Figure 5A–E). In addition, the phosphorylation of p65NF-κB and expression of MUC5AC was significantly increased in the lung tissues from the OVA-challenged mice (Figure 5F,G). In contrast, LOL treatment significantly suppressed the phosphorylation of MAPKs/AP-1, p65NF-κB, and expression of MUC5AC in the lung tissues compared with the OVA-challenged asthma model.

### 3.6. Effects of LOL on Nrf-2 Pathway, Productions of ROS and NO, and Oxidative Stress Markers in the Lung Tissue of the OVA-Challenged Asthma Model

OVA-challenged mice exhibited a significant decrease in nuclear Nrf-2, HO-1, and NQO-1 expression compared to the normal controls (Figure 6A–C). OVA-challenged mice showed an increase in levels of ROS, NO, and TBARS, as well as a decrease in activities of GSH, CAT, and SOD compared with normal controls (Figure 6D–I). In contrast, LOL treatment up-regulated nuclear translocation of Nrf-2, HO-1, and NQO1 expression compared with the OVA-challenged mice. Moreover, LOL treatment suppressed the productions of ROS, NO, and TBARS, whereas elevated activities of GSH, CAT, and SOD in lung tissues compared with the OVA-challenged mice.

### 3.7. Effects of LOL on Pro-inflammatory Cytokines and Th2 Cytokines Production in TNF-α-Stimulated NCI-H292 Cells

Based on the results of cytotoxicity analysis, nontoxic concentrations (25, 50, and 100 μg/mL) of LOL were employed in the present study (Figure 7A). TNF-α-stimulated cells markedly increased the production of pro-inflammatory cytokines, such as TNF-α and IL-6. However, LOL-treated cells had a significant decrease in the levels of TNF-α and IL-6 in a concentration-dependent manner (Figure 7B,C). TNF-α-stimulated cells up-regulated the mRNA levels of IL-4, IL-5, and IL-13 compared with the non-treated cells, whereas LOL-treated cells reduced the mRNA levels of Th2 cytokines (Figure 7D–E).

### 3.8. Effects of LOL on P65NF-κB and Nrf-2 Pathway, ROS Production, Oxidaive Stress Markers and DPPH Radical Scavenging Activity in TNF-α-Stimulated NCI-H292 Cells

As presented in Figure 8, TNF-α treatment increased the phosphorylation of p65NF-κB and down-regulated the nuclear translocation of Nrf-2 and HO-1 (Figure 8A–C). TNF-α treatment increased the ROS production and decreased the GSH content and SOD activity in NCI-H292 cells (Figure 8D–F). In contrast, LOL-treated cells inhibited the phosphorylation of p65NF-κB compared with the TNF-α-stimulated cells. LOL treatment increased the translocation of Nrf-2 into the nucleus with elevated HO-1 expression. LOL markedly suppressed ROS production, and increased GSH contents, as well as SOD activities. In addition, LOL exhibited a significant increase in DPPH radical scavenging activity in a concentration-dependent manner (Figure G).

## 4. Discussion

Recently, many researchers highlighted the usefulness of anti-asthmatic herbal medicine for the treatment of patients with moderate-severe allergic asthma [49,50]. Therefore, novel approaches for asthma therapy have been developed with herbal remedies as a form of complementary or alternative medicine. In our study, we evaluated the therapeutic effects of LOL on OVA-challenged asthma mice and TNF-α-stimulated NCI-H292 cells. LOL treatment up-regulated nuclear translocation of Nrf-2, HO-1 expression, and levels of GSH and SOD, and inhibited ROS production, pro-inflammatory cytokines (TNF-α, IL-4, IL-5, IL-6, and IL-13) in TNF-α-stimulated NCI-H292 cells. In the OVA-challenged mice, LOL treatment attenuated inflammatory cell infiltration, mucus hypersecretion in lung tissues, and airway hyper-responsiveness. LOL also significantly reduced the levels Th2 cytokines, MUC5AC, and eotaxin in BALF, and total IgE and OVA-specific IgE levels in serum. LOL treatment effectively inhibited the activation of MAPKs/AP-1 and p65NF-κB in the OVA-challenged asthma mice model. In addition, LOL treatment markedly enhanced Nrf-2/HO-1/NQO1 signaling with elevated the activities of anti-oxidant enzymes, such as GSH, CAT, and SOD, and suppressed the levels of ROS, NO, and TBARS in lung tissues of the OVA-challenged allergic asthma model.

Allergic asthma is defined as a chronic inflammatory condition of the airway. It is characterized by airway eosinophilia, goblet cell hyperplasia, and hyper-responsiveness in response to inhaled allergens or non-specific stimuli [5]. These phenomena are accompanied by Th2 cell activation and release of various cytokines (IL-4, IL-5, and IL-13) and chemokines (eotaxin), which are involved in the development of asthma [5,51]. Th2 cytokines induce IgE switching in B lymphocytes, mucus hypersecretion from goblet cells, and recruitment of eosinophils in the airway [52]. The eotaxin released from Th2 cells also aggravates the recruitment of eosinophils into the airway [53]. Therefore, modulation of the Th2 cytokine might be an important factor for the treatment of asthma [54]. In this study, LOL treatment decreased the production of Th2 cytokines, eotaxin in the BALF of OVA-challenged mice and down-regulated the IL-4, IL-5, and IL-13 mRNA expression levels in TNF-α-stimulated NCI-H292 cells. These results are consistent with the analysis of lung histology and airway hyper-responsiveness. LOL-treated mice exhibited attenuated airway inflammation, mucus production, and airway hyper-responsiveness. These findings indicate that LOL treatment effectively suppresses asthmatic responses in OVA-challenged mice by inhibiting airway inflammation with reduction of Th2 cytokines.

Oxidative stress is caused by the imbalance between the ability of ROS formation and capacity of the anti-oxidative defense system that causes destructive damage to a variety of tissues [55]. Accumulating evidence suggests that ROS-mediated oxidative stress plays a central role in the pathogenesis in allergic asthma [2,55]. Nrf-2 signaling is one of the most important anti-oxidative stress mechanisms in the cellular defense system [12]. Counteract to excessive production of ROS, Nrf-2 is separated from Kelch-like ECH-associated protein 1, and translocated from the cytosol into the nucleus for binding to ARE region, result in the elevation of HO-1 and NQO1 expressions [55,56]. The activation of the Nrf-2 pathway suppresses oxidative stress and ROS-mediated inflammatory responses [57]. In this study, LOL treatment significantly increased translocation of Nrf-2 to nucleus, and elevated HO-1 and NQO1 levels in OVA-challenged mice and TNF-α-stimulated NCI-H292 cells. LOL effectively inhibited the production of ROS, NO, and level of TBARS, whereas restored the activities of anti-oxidant enzymes, such as GSH, CAT, and SOD in OVA-challenged mice and TNF-α-stimulated NCI-H292 cells. Moreover, LOL exhibited a potent DPPH radical scavenging activity. Therefore, these results indicate that anti-allergic effects of LOL on OVA-challenged allergic asthma is closely associated with activating of Nrf-2/HO-1/NQO1 and its anti-oxidant activities.

ROS can activate MAPKs, which play proliferation of inflammatory cells, differentiation of Th2 cells, and the production of inflammatory cytokines in allergic asthma [58]. The phosphorylation of MAPKs is found to induce the activation of AP-1, such as c-Jun, and c-Fos, which produce MUC5AC and Th2 cytokine in the OVA-challenged allergic asthma model [16,59]. On the other hand, ROS promotes the activation of NF-κB, which causes the progression of allergic inflammation in asthmatic patients [60]. In the NF-κB signaling pathway, IκB is phosphorylated by IκB kinase and then NF-κB dissociated from IκB bind to DNA in the promoter region of the target gene as a dimer, NF-κB1 (p50) and RelA (p65), to regulate the expression of inflammatory cytokines and MUC5AC [61,62]. MUC5AC expression is increased in goblet cells, leading to airway limitation in allergic asthma [62]. In this study, LOL treatment suppressed the activation of MAPKs/AP-1 and p65NF-κB, and inhibited expression of MUC5AC in OVA-challenged asthma mice. The down-regulation of MAPKs/AP-1 and p65NF-κB may be related to the suppression of MUC5AC in asthmatic lung tissues of OVA-challenged mice. These results indicate that LOL attenuated allergic responses by inhibiting MAPKs/AP-1 and NF-κB in OVA-challenged asthma mice.

*L. obtusiloba*, a traditional Korean medicine, is used to treat inflammation and improve blood circulation [33]. *L. obtusiloba* is reported to contain actifolin, pluviatilol, (+)-syringaresinol, and (+)-9’-0-trans-feruloyl-5,5’-dimethoxylariciresinol [63]. The beneficial effects of *L. obtusiloba* include its anti-oxidant and anticancer activities, and its ability to inhibit histamine release in mast cells [34,36,64]. According to previous studies, the anti-inflammatory effect of *L. obtusiloba* may be due to its ability to suppress the pro-inflammatory cytokines via the inhibition of the NF-κB activation [36]. On the other hand, the anti-oxidant properties of *L. obtusiloba* include scavenging DPPH radicals, reduction of TBARS concentration and elevation of GSH concentration in *tert*-butyl hydroperoxide-induced HepG2 cells and rats. In addition, the 70% ethanol extract of *L. obtusiloba* bloom contains two major compounds, quercitrin and kaempferol rhamnoside (afzelin) [48]. In this study, quercetin rhamnoside (quercitrin; 2.60%) and kaempferol rhamnoside (afzelin; 1.58%) are the major identified components of LOL. Quercetin decreases TBARS concentration and iNOS expression and increases superoxide dismutase in a rat model of aspiration-induced lung injury [65]. Kaempferol rhamnoside suppresses ROS production and Th2 cytokine, and elevates DPPH scavenging activity in a mouse model of allergic asthma [66]. However, there has been no report on the anti-allergic asthma effects of LOL in an in vivo model, as well as its underlying mechanism. Based on our findings, LOL effectively attenuated airway inflammation and oxidative stress by suppressing of MAPKs and p65NF-κB activation, and by activating of Nrf-2/HO-1/NQO1 signaling in the OVA-challenged asthma model.

Collectively, these results demonstrated the protective effects of LOL on allergic asthma. However, there is some limitations in our study. It is yet not known which compounds of LOL possess the anti-oxidant and anti-inflammatory activities in the OVA-challenged asthma model. Hence, it is required to clarify the contents and activities of the two major active components for further therapeutic application on allergic asthma.

## 5. Conclusions

This is the first study to demonstrate that treatment with LOL effectively attenuates airway inflammation, mucus hypersecretion, and ROS-mediated oxidative stress in the OVA-challenged allergic asthma model and TNF-α-stimulated NCI-H292 cell. These anti-asthmatic effects of LOL may be due to its ability to induce anti-oxidant activities through activation of Nrf-2/HO-1/NQO1, and inhibit inflammatory mediator and mucus production by suppressing of MAPKs/p65NF-κB. Thus, our results suggest that LOL may be a useful therapeutic agent against allergic asthma.

## Figures and Tables

**Figure 1 antioxidants-09-00563-f001:**
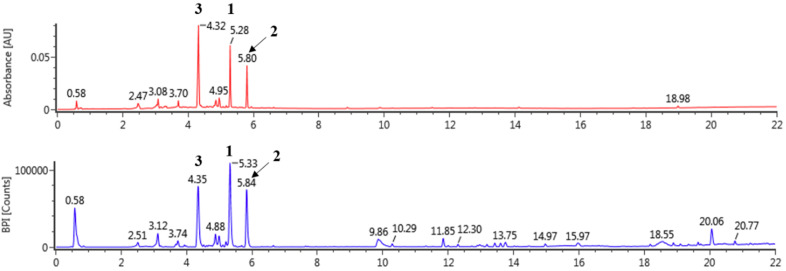
The UPLC-Q-ToF-MS BPI chromatogram of *Lindera obtusiloba* in negative mode.

**Figure 2 antioxidants-09-00563-f002:**
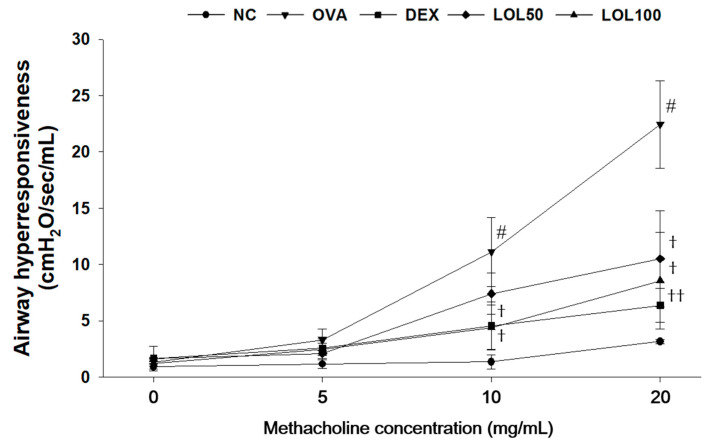
Effects of LOL on airway hyper-responsiveness in OVA-challenged asthma model. NC: normal control mice; OVA: OVA-challenged mice; DEX: dexamethasone (3 mg/kg) + OVA-challenged mice; LOL: LOL (50 or 100 mg/kg) + OVA-challenged mice. The values are expressed as the means ± SD (*n* = 7/group). ^#^
*p* < 0.01, significantly different from NC group; ^†, ††^
*p* < 0.05, *p* < 0.01, significantly different from OVA-challenged group.

**Figure 3 antioxidants-09-00563-f003:**
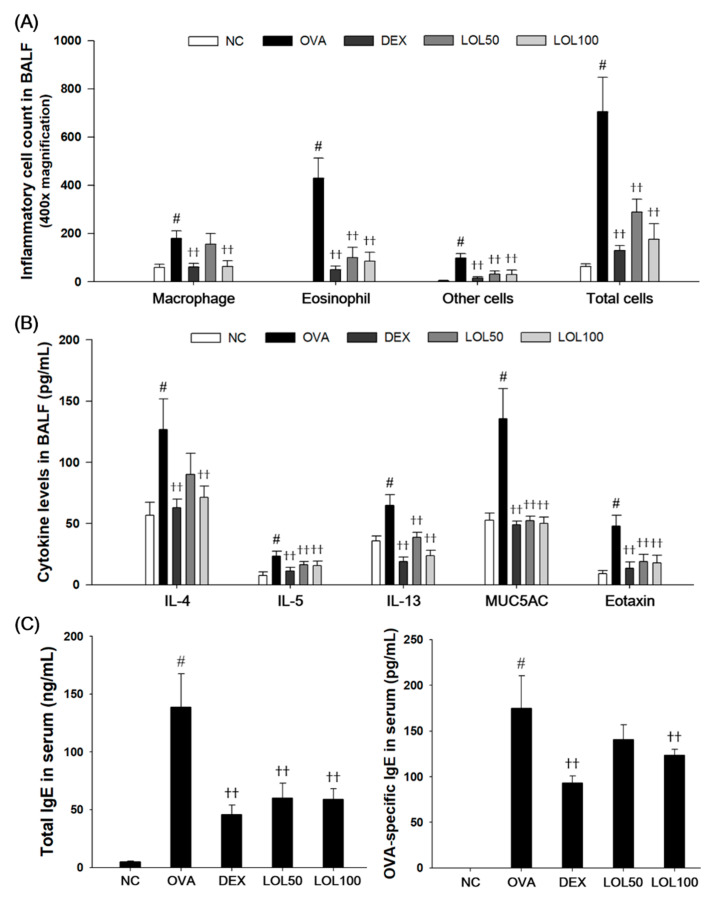
Effects of LOL on inflammatory cell number, Th2 cytokines, eotaxin, MUC5AC, total IgE and OVA-specific IgE in OVA-challenged asthma model. (**A**) The inflammatory cells were attached on slide and stained with Diff-Quik stain reagent. (**B**) The levels of IL-4, IL-5, IL-13, eotaxin, and MUC5AC in the BALF were determined by ELISA. (**C**) The total IgE and OVA-specific IgE in the serum were determined using ELISA. NC: normal control mice; OVA: OVA-challenged mice; DEX: dexamethasone (3 mg/kg) + OVA-challenged mice; LOL: LOL (50 or 100 mg/kg) + OVA-challenged mice. The values are expressed as the means ± SD (*n* = 7/group). ^#^
*p* < 0.01, significantly different from NC group; ^††^
*p* < 0.01, significantly different from OVA-challenged group.

**Figure 4 antioxidants-09-00563-f004:**
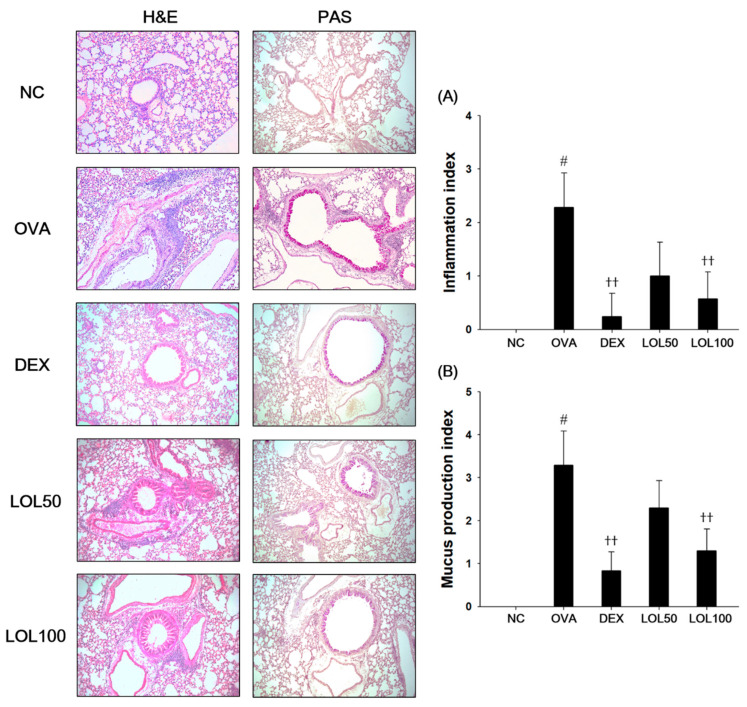
Effects of LOL on airway inflammation and mucus production in OVA-challenged asthma model. Histopathological analysis on (**A**) airway inflammation and (**B**) mucus production was performed in the lung tissues by hematoxylin and eosin (H&E, left panel) and Periodic acid-Schiff (PAS, right panel) staining. NC: normal control mice; OVA: OVA-challenged asthma mice; DEX: dexamethasone (3 mg/kg) + OVA-challenged asthma mice; LOL: LOL (50 or 100 mg/kg) + OVA-challenged asthma mice. The values are averaged pathologic scores of inflammatory cell infiltration and mucus production. The values expressed as the means ± SD (*n* = 7/group). ^#^
*p* < 0.01, significantly different from NC group; ^††^
*p* < 0.01, significantly different from OVA-challenged group.

**Figure 5 antioxidants-09-00563-f005:**
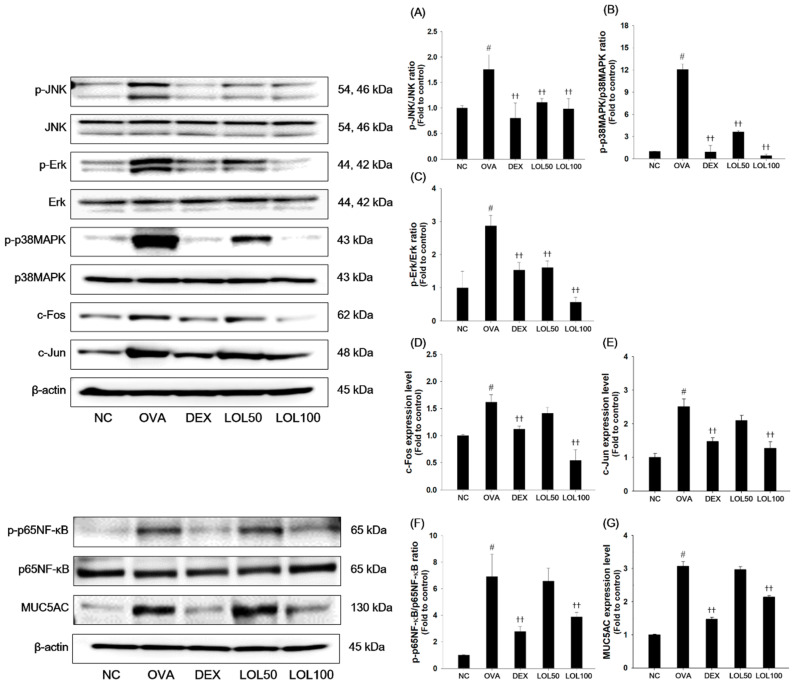
Effects of LOL on activation of MAPKs/AP-1, p65NF-κB and expression of MUC5AC in OVA-challenged asthma model. The protein levels of (**A**–**E**) MAPKs/AP-1, (**F**) p65NF-κB and (**G**) MUC5AC in the lung tissues were determined by western blot analysis. β-actin was used to confirm equal protein loading. NC: normal control mice; OVA: OVA-challenged asthma mice; DEX: dexamethasone (3 mg/kg) + OVA-challenged asthma mice; LOL: LOL (50 or 100 mg/kg) + OVA-challenged asthma mice. The values are expressed as the means ± SD (*n* = 7/group). ^#^
*p* < 0.01, significantly different from NC group; ^††^
*p* < 0.01, significantly different from OVA-challenged group.

**Figure 6 antioxidants-09-00563-f006:**
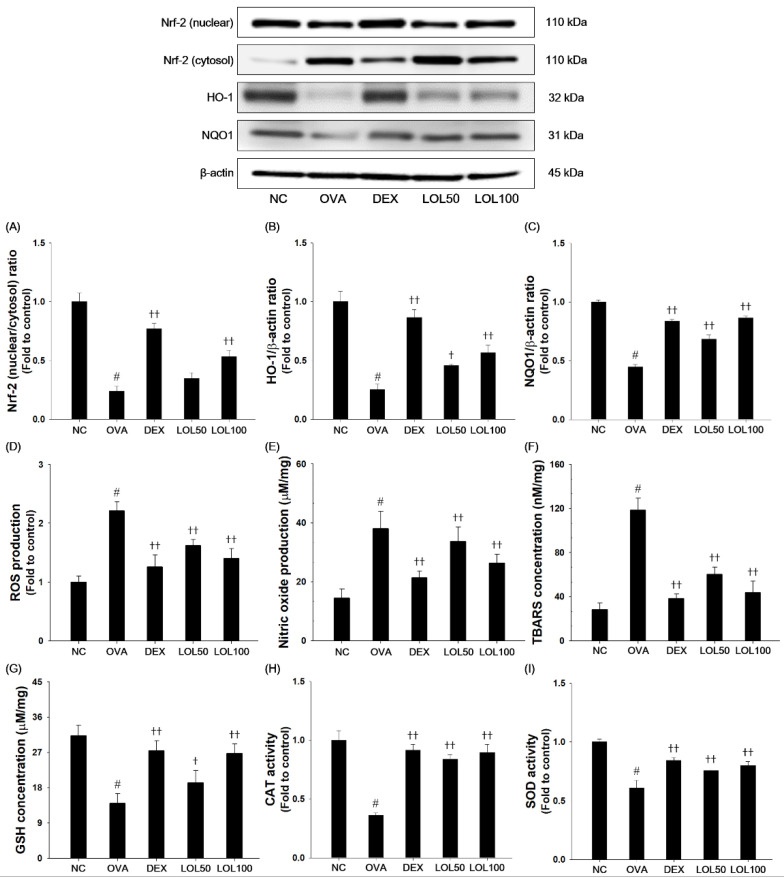
Effects of LOL on OVA-challenged Nrf-2 pathway and oxidative stress. The (**A**) Nrf-2, (**B**) HO-1 and (**C**) NQO1 in the lung tissues were analyzed by western blot (β-actin: loading control). The levels of (**D**) ROS and (**E**) NO, contents of (**F**) TBARS and (**G**) GSH, and activities of (**H**) CAT and (**I**) SOD were determined in the lung tissues. NC: normal control mice; OVA: OVA-challenged asthma mice; DEX: dexamethasone (3 mg/kg) + OVA-challenged asthma mice; LOL: LOL (50 or 100 mg/kg) + OVA-challenged asthma mice. The values are expressed as the means ± SD (*n* = 7/group). ^#^
*P* < 0.01, significantly different from NC group; ^†,††^
*P* < 0.05, *P* < 0.01, significantly different from OVA-challenged group.

**Figure 7 antioxidants-09-00563-f007:**
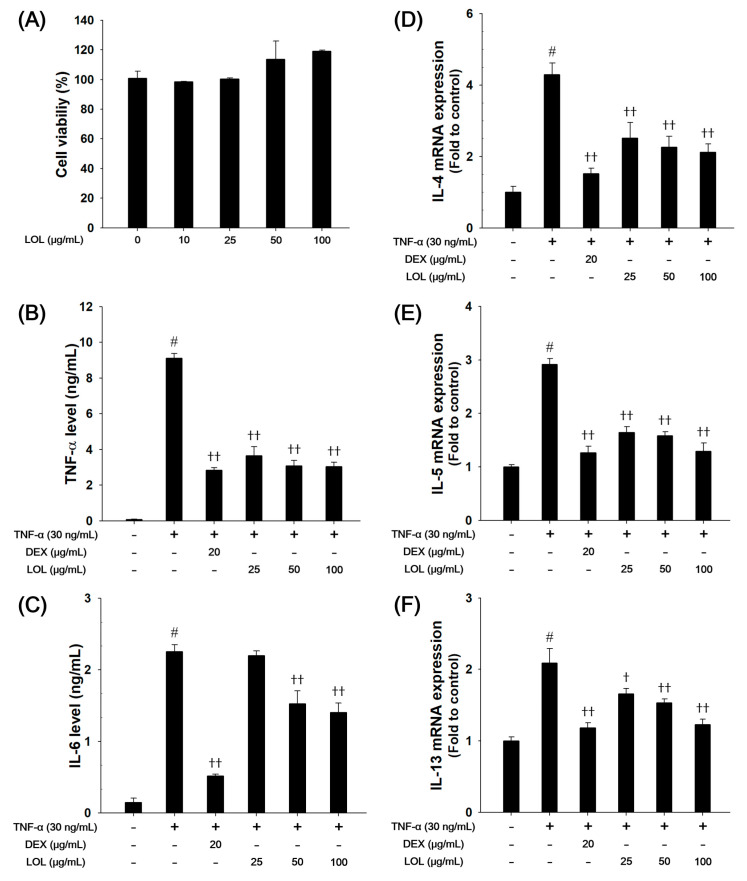
Effects of LOL on cell viability, pro-inflammatory cytokines and Th2 cytokines, including IL-4, IL-5 and IL-13, in TNF-α-stimulated NCI-H292 cells. The (**A**) cell viability was measured using a WST-1 reagent, and LOL was treated with 10, 25, 50, and 100 μg/mL for 24 h. The levels of (**B**) TNF-α and (**C**) IL-6 were determined by ELISA. The culture medium were changed RPMI1640 (0.1% FBS) and treated with LOL (25, 50 and 100 μg/mL) and DEX (20 μg/mL) for 1 h and incubated with TNF-α (30 ng/mL) for 30 min (TNF-α) or 24 h (IL-6). The mRNA levels of (**D**) IL-4, (**E**) IL-5, and (**F**) IL-13 were measured by real-time PCR in TNF-α-stimulated NCI-H292 cells. Cells were treated with LOL (25, 50 and 100 μg/mL) and DEX (20 μg/mL) for 1 h and incubated with TNF-α (30 ng/mL) for 18 h. The values are expressed as the means ± SD (*n* = 3). ^#^
*p* < 0.01, significantly different from control; ^†,††^
*p* < 0.05, *p* < 0.01, significantly different from TNF-α-treated group.

**Figure 8 antioxidants-09-00563-f008:**
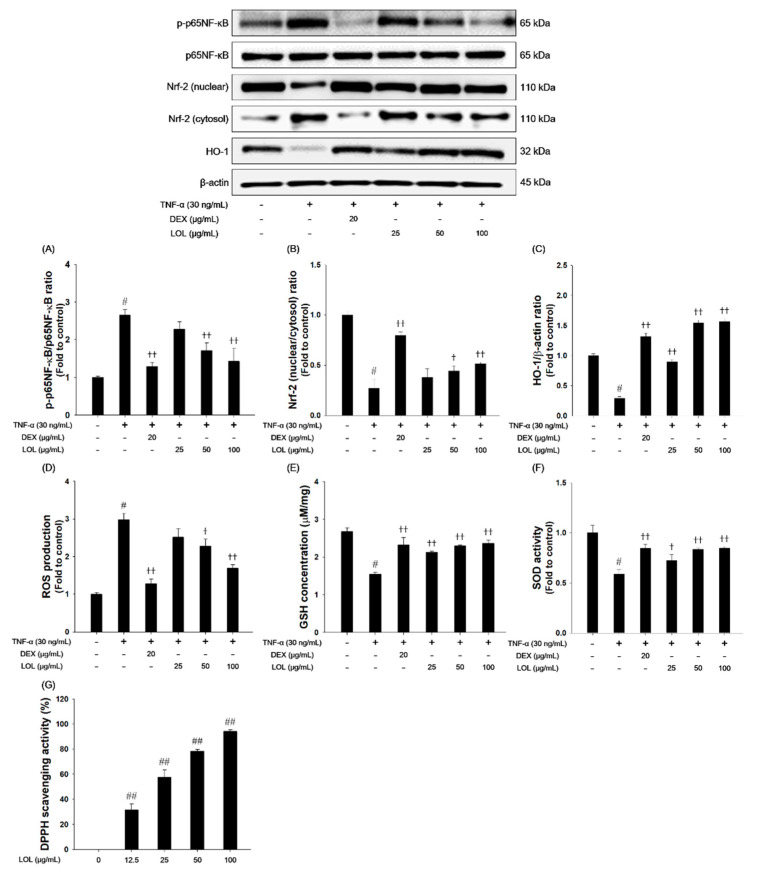
Effects of LOL on p65NF-κB, Nrf-2, HO-1 and oxidative stress makers in TNF-α-stimulated NCI-H292 cells. The (**A**) p-p65NF-κB, (**B**) Nrf-2 and (**C**) HO-1 were analyzed by western blot (β-actin: loading control). The levels of (**D**) ROS and (**E**) GSH, and activities of (**F**) SOD were determined in TNF-α-stimulated NCI-H292 cells. The cells treated with LOL (25, 50 and 100 μg/mL) and DEX (20 μg/mL) for 1 h and incubated with TNF-α (30 ng/mL) for 30 min (p-p65NF-κB, Nrf-2, HO-1, GSH and SOD) or 24 h (ROS production). (**G**) The DPPH radical scavenging activity was evaluated by DPPH assay. Each value represents the means ± SD (*n* = 3). ^#^
*p* < 0.01 and ^##^
*p* < 0.001, significantly different from control; ^†^
*p* < 0.05, ^††^
*p* < 0.01, significantly different from TNF-α-treated group.

**Table 1 antioxidants-09-00563-t001:** Tentatively identification of major peaks detected in *Lindera obtusiloba.*

NO.	t*_R_* (min)	Formula	Detected *m*/*z*	Exacted m/z	Error (ppm)	Fragments	Identification	Content (mg/g)
1	5.33	C_21_H_20_O_11_	447.09323	447.09329	0.12	447, 301, 300, 271,255, 243	Quercetin rhamnoside	26.04 ± 0.05
2	5.84	C_21_H_20_O_10_	431.09840	431.09837	0.07	431, 285	Kaempferol rhamnoside	15.82 ± 0.02
3	4.35	C_20_H_20_O_11_	435.09433	435.09329	1.26	435, 303, 285, 151	Unknown	-

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
