# Peer review of "Lindera obtusiloba Attenuates Oxidative Stress and Airway Inflammation in a Murine Model of Ovalbumin-Challenged Asthma"

_antioxidants, 2020, doi:10.3390/antiox9070563_

Round 1
Reviewer 1 Report
- Please explain the calculation method of inflammation index and mucus index in the “Materials and Methods” section.
- Please explain the nuclear separation method in the “Materials and Methods” section.
- Please replace the DPPH method and result with the data of ROS levels in cell experiment.
- Please add the data of the activities of SOD and catalase in vivo and in vitro.
- Please explain the similarities and differences between the result of reference 41 and your result. Were the components of 70% ethanolic extract same as the methanol extract?
- Please explain whether the pharmacological activities of flavonoid rhamnoside and flavonoids will be the same in vivo and in vitro?
- TNF-alpha was used as an inducer in vitro experiment. Why? Please explain relationship between TNF-alpha and OVA.
- The positive control was lack in vitro experiment.
- The content of three components in LOL was lack.
Author Response
1. Please explain the calculation method of inflammation index and mucus index in the “Materials and Methods” section.
It has been clarified in the Materials and Methods section as follow:
Page 4, L171-174.
The degree of inflammation and mucus production of each slide were graded as follows: 0, no lesions; 1, minimal; 2, mild; 3, moderate; and, 4, severe. The index levels of inflammation and mucus production were represented as mean ± SD.
2. Please explain the nuclear separation method in the “Materials and Methods” section.
It has been described in the Materials and Methods:
Page 4, L176-L183
The lung tissues from mice were homogenized (1/10, w/v) using a homogenizer in CER I (NE-PER Nuclear and Cytoplasmic Extraction Reagent, Thermo Scientific). The supernatants of tissue were mixed with ice-cold CER II and incubated at 1 min. After incubation, samples were vortexed and centrifuged at 16000 × g for 10 min. The supernatants was collected (cytoplasmic extract) and mixed with NER and vortexed for 15 sec every 10 min, for a total of 40 min. The samples were centrifuged at 16000 × g for 10 min and collected supernatants (nuclear fraction). The cytoplasmic and nuclear fractions were determined using a Bradford reagent (Bio-Rad Laboratories).
3. Please replace the DPPH method and result with the data of ROS levels in cell experiment.
It has been revised in the Materials and Methods of DPPH radical scavenging activity and ROS production as follow:
Please see the attachment.
Page 5, L195-204.
2.9. Measurement of Nitric Oxide (NO) and ROS Production
The lung tissues were grinded using a homogenizer in ice-cold PBS (1/10, w/v). The homogenized lung tissues were centrifuged at 11000 × g for 15 min at 4°C and collected to supernatants. The supernatants were used for measurement to nitric oxide using nitric oxide plus kit (iNtRoN Biotechnology, Republic of Korea). The absorbance at 550 nm was measured using microplate reader (Bio-Rad Laboratories).
The ROS production was monitored using CellRox® green reagent (Life technologies, NY, USA), which could be used to evaluate ROS production. The samples were incubated with CellRox® green reagent (5 μM) for 30 min at 37°C. After incubated, ROS activity was measured by measuring the fluorescence at 485 nm excitation and 520 nm emission on a fluorescence plate reader (Perkin-Elmer, MA, USA).
Page 6, L239-246.
2.11. Measurement of 2,2-Diphenyl-1-Picryl Hydrazyl (DPPH) Radical Scavenging Activity
The DPPH (Sigma-Aldrich) radical scavenging assay of LOL was performed as described previously with Abderrahim et al. [37]. Various concentrations (0, 12.5, 25, 50, and 100 μg/mL, 100 μL) of LOL were mixed with 200 μM DPPH radical solution (dissolved in methanol, 100 μL) in 96-well microplate at room temperature for 30 min. The absorbance was measure at 520 nm with a blank containing DPPH and methanol. The DPPH radical scavenging activity (%) was calculated with the following equation: (1 − (As/Ac)) × 100, where As is the absorbance in the present of sample and Ac is the absorbance in the absence of sample.
In addition, the results of ROS level has been added in Figure 6 and Figure 8 with Results sections.
Page 11, L378-L386.
3.6. Effects of LOL on Nrf-2 Pathway, production of ROS and NO, and Oxidative Stress Markers in the Lung Tissue of the OVA-challenged Asthma Model
As presented in Figure 6, OVA-challenged mice showed a significant decrease in nuclear Nrf-2, HO-1, and NQO-1 expression compared to the normal controls. OVA-challenged mice exhibited an increase in levels of ROS, NO, and TBARS, as well as a decrease in activities of GSH, CAT, and SOD compared with normal controls. In contrast, LOL treatment upregulated nuclear translocation of Nrf-2, HO-1, and NQO1 expression compared with the OVA-challenged mice. Moreover, LOL treatment suppressed the productions of ROS, NO, and TBARS, whereas elevated activities of GSH, CAT, and SOD in lung tissues compared with the OVA-challenged mice.
Page 13, L415-Page 14, L424.
3.8. Effects of LOL on p65NF-κB and Nrf-2 Pathway, Oxidative Stress and DPPH Radical scavenging activity in TNF-α-stimulated NCI-H292 Cells
As shown in Figure 8, TNF-α-stimulated cells increased the phosphorylation of p65NF-κB and decreased the nuclear translocation of Nrf-2 and HO-1. In addition, TNF-α treatment increased the ROS production and decreased the GSH content and SOD activity in NCI-H292 cells. In contrast, LOL-treated cells inhibited the phosphorylation of p65NF-κB compared with the TNF-α-stimulated cells. LOL treatment increased the translocation of Nrf-2 into the nucleus with elevated HO-1 expression. LOL markedly suppressed ROS production, and increased GSH contents, as well as SOD activities. In addition, LOL exhibited a significant increase in DPPH radical scavenging activity in a concentration-dependent manner.
4. Please add the data of the activities of SOD and catalase in vivo and in vitro.
The analysis methods of SOD and catalase activities have been added in the Materials and Methods sections and the results of SOD and catalase were presented in Figure 6. (in vivo) and Figure 8. (in vitro), respectively, as described above :
Please see the attachment.
Page 5, L220-L227.
2.10. Oxidative Stress Markers Analysis
The SOD activity (In in the lung tissues were determined using the EZ-SOD kit, lung samples (20 μL) were mixed with WST working solution (200 μL) and incubated with enzyme working solution at 37°C for 20 min. The absorbance was determined as 450 nm. The SOD activity was normalized to the value of the control.
The CAT activity was measured by quantifying the fluorescence at 540 nm (excitation)/590 nm (emission). The Samples were added to hydrogen peroxide (40 μM) in reaction buffer (25 μL) and incubated for 30 min. After incubation, each samples was incubated with oxi-probe/HRP working solution at 37°C for 30 min. The CAT activity was normalized to the value of the control.
5. Please explain the similarities and differences between the result of reference 41 and your result. Were the components of 70% ethanolic extract same as the methanol extract?
It has been revised in the Discussion as follow:
Page 16, L498-L501.
In addition, the 70% ethanol extract of L. obtusiloba blume contain two major of compound, such as quercitrin and kaempferol rhamnoside (afzelin) [44]. However, in this study, taxifolin pentoside, quercetin rhamnoside (quercitrin) and kaempferol rhamnoside (afzelin) are the major identified components of methanol of LOL.
- Hong C.O., Rhee C.H., Won N.H., Choi H.D., Lee K.W. Protective effect of 70% ethanolic extract of Lindera obtusiloba Blume on tert-butyl hydroperoxide-induced oxidative hepatotoxicity in rats. Food Chemical Toxicology. 53:214–220 (2013).
6. Please explain whether the pharmacological activities of flavonoid rhamnoside and flavonoids will be the same in vivo and in vitro?
Previous studies demonstrated that quercetin and quercetin rhamnoside (quercitrin) have potent anti-inflammatory response in vitro and in vivo (Dai et al., 2013; Monica et al., 2005). The anti-inflammatory effects of quercetin and quercetin rhamnoside inhibit production of ROS and nitric oxide, and decrease expression of iNOS through suppression of NF-κB signaling in cytokine-induced injuries of RINm5F β-cells. In addition, the quercetin and quercetin rhamnoside attenuate pro-inflammatory cytokine and inducible nitric oxide through downregulation of NF-κB in dextran sulfate sodium-induced IBD model. Therefore, quercetin and quercetin rhamnoside have a similar activity of anti-inflammation in vitro and in vivo.
Dai X., Ding Y., Zhang Z., Cai X., Li Yong. Quercetin and quercitrin protect against cytokine-induced injuries in RINm5F β-cells via mitochondrial pathway and NF-κB signaling. International Journal of Molecular Medicine 31:265-271 (2013)
Monica C., Desiree C., Saleta S., Isabel B., Jordi X., Julio G., Antonio Z. In vivo quercitrin anti-inflammatory effect involves release of quercetin, which inhibits inflammation through down-regulation of the NF-κB pathway. European Journal of immunology. 35(2):584-592 (2005).
7. TNF-alpha was used as an inducer in vitro experiment. Why? Please explain relationship between TNF-alpha and OVA.
The relationship between of TNF-alpha and OVA have been described in Materials and Methods section as follow:
Page 6, L249-L251.
To further therapeutic application of LOL for asthma treatment, TNF-α-stimulated human lung epithelial cells, NCI-H292, has been used to evaluate the anti-inflammatory effects in vitro corresponding to OVA-challenged asthma mice model [38-40].
- Kawamoto Y., Morinaga Y., Kimura Y., Kaku N., Kosai K., Uno N., Hasegawa H., Yanagihara K. TNF-alpha inhibits the growth of Legionella pneumophila in airway epithelial cells by inducing apoptosis. Journal of Infection and Chemotherapy. 23:51–55 (2017).
- Ryu E.K., Kim T.H., Jang E.J., Choi Y.S., Kim S.T., Hahm K.B. Wogonin, a plant flavone from Scutellariae radix, attenuated ovalbumin-induced airway inflammation in mouse model of asthma via the suppression of IL-4/STAT6 signaling. Journal of Clinical Biochemistry and Nutrition. 57(2):105-112 (2015).
- Kurakula K., Vos M., Logiantara A., Roelofs J.J., Nieuwenhum M.A., Koppelman G.H., Postma D.S., Van Rijt L.S., De Vries C.J. Nuclear receptor Nur 77 attenuates airway inflammation in mice by suppressing NF-κB activity in lung epithelial cells. The Journal of Immunology. 195(4):1388-1398 (2015).
8. The positive control was lack in vitro experiment.
Unfortunately, we didn’t used positive control in vitro experiment. Further study using main components of LOL, we will apply the positive control to compare the activities of these components as you pointed out. This point has been described in Discussion section as below.
9. The content of three components in LOL was lack.
We really agreed with your opinion. The content analysis of three components need to further application and precise comprehension of L.obtusiloba's biological activity. Regretfully, we did not conduct the content analysis of major components of L.obtusiloba. However, we plan the analysis of contents of three component in L. obtusiloba and, if possible, we should conduct and add the analysis results as supplement data prior to publication or further study using three components (about 3 or 4 weeks need). This point also has been described in Discussion section as follow:
Page 16, L509-515.
Collectively, these results demonstrated the protective effects of LOL on allergic asthma. However, there is some limitations in this study. Firstly, the effects of LOL did not compare with positive control in vitro experiments. For precise comprehension of L. obtusiloba’s biological activities, further investigation applies to positive control in vitro. Secondly, it is yet not known which components possess the anti-inflammatory/anti-oxidant activities in the OVA-challenged asthma model. Hence, it is required to clarify the contents and activities of the three major active components for further therapeutic application on allergic asthma.

Reviewer 2 Report
The present pare is very interesting as it reports the bioactivity of Lindera obtusiloba against asthma of murine model systems. I have some suggestions to improve the paper before its publication.
a) in the title the name of the species should be not in capital.
b) in the introduction, the authors should report that plant extracts are prone to modulate animal cell oxidative status because it is minimally mentioned. Moreover, the concept of phytocomplex should be reported. I suggest to support such declarations with the following citations: Phytotherapy Research, 2010, 24(4), 595-601; Phytomedecine, Volume 46, 2018, Pages 1-10; Pharmacognosy research, 2016, 8.Suppl 1: S42; European Journal of Pharmaceutical Sciences, 2017, 96, 53-61.
c) why only female animals were selected?
d) How cDNA was synthesized?
e) Where did the authors take primer sequences for qPCR?
f) It is not clear if the authors also analyzed pure standards in UPLC. If so, could they show a profile with only the standard molecules? g) ROS and RNS should be also measured in this work, to better confirm their data and the hypothesis of redox state modulation h) a graphical representation which summarizes the main data obtained by the present work should be added, to collect all together the effects of the plant extract in mice
Author Response
Response to Reviewer’s comments
The present pare is very interesting as it reports the bioactivity of Lindera obtusiloba against asthma of murine model systems. I have some suggestions to improve the paper before its publication.
a) in the title the name of the species should be not in capital.
It has been revised in the Title; Lindera obtusiloba Attenuates Oxidative Stress and Airway Inflammation in a Murine Model of Ovalbumin-challenged Asthma
b) in the introduction, the authors should report that plant extracts are prone to modulate animal cell oxidative status because it is minimally Moreover, the concept of phytocomplex should be reported. I suggest to support such declarations with the following citations: Phytotherapy Research, 2010, 24(4), 595-601; Phytomedecine, Volume 46, 2018, Pages 1-10; Pharmacognosy research, 2016, 8.Suppl 1: S42; European Journal of Pharmaceutical Sciences, 2017, 96, 53-61.
It has been added in the Introduction as follow:
Page 2, L81-L87.
Previous studies demonstrated that the extract of L. obtusiloba has potent anti-oxidative and anti-inflammatory activities on atopic dermatitis and anti-allergic responses in mast cells [29-32]. In detail, L. obtusiloba has effects such as anti-inflammation and anti-allergy through the suppression of histamines and Th2 cytokines in a DFE/DNCB-induced atopic dermatitis model [30]. On the other hand, L. obtusiloba inhibited ROS generation in the Fe3+-EDTA/H2O2 system, and has potent DPPH radical scavenging activity [31]. However, there is no reported evidence for the anti-inflammatory effects of L. obtusiloba leaves (LOL) on allergic asthma in vivo.
c) why only female animals were selected?
It has been clarified in the Materials and Methods section as follow:
Page 3, L118-L120.
Previous study demonstrated that female mice are more sensitive to development of allergic airway inflammation than male mice [33]. Thus, we used female BALB/c mice to develop the OVA-challenged asthma model based on previous studies [2,33]
2. Shin I.S., Hong J., Jeon C.M., Shin N.R., Kwon O.K., Kim H.S., Kim J.C., Oh S.R., Ahn K.S. Diallyl-disulfide, an organosulfur compound of garlic, attenuates airway inflammation via activation of the Nrf-2/HO-1 pathway and NF-kappaB suppression. Food and Chemical Toxicology. 62:506–513 (2013).
33. Melgert B.N., Postma D.S., Kuipers I., Geerlings M., Luinge M.A., Van Der Strate B.W., Kerstjens H.A., Timens W., Hylkema M.N. Female mice are more susceptible to the development of allergic airway inflammation than male mice. Clinical and Experimental Allergy. 35(11):1496-1503 (2005).
d) How cDNA was synthesized?
It has been clarified in the Materials and Methods section as follow:
Page 6, L267-L272.
The cells (5 × 104 cells/well) were incubated with LOL (0, 25, 50, and 100 μg/mL) followed by TNF-α (30 ng/mL) for 18 h. Thereafter, the cells were washed twice with PBS and total RNA was isolated using an RNA extraction kit (RNeasy®; Qiagen, Valencia, CA, USA). The RNA concentration and purity were measured microphotometer (Allsheng Instruments Co., Hangzhou, China). For real time PCR, total RNA (1 μg) was used for cDNA synthesis, employing iScript cDNA synthesis kit (Bio-Rad Laboratories).
e) Where did the authors take primer sequences for qPCR?
It has been added Gene-Bank accession number as follow:
Page 6, L275-L282.
IL-4 forward; 5’- ATG GGT CTC ACC TCC CAA CT -3’, IL-4 reverse; 5’- TAT CGC ACT TGT GTC CGT GG-3’ (Gene-Bank accession number: NM_172348.3), IL-5 forward; 5’- CAG GGA ATA GGC ACA CTG GA -3’, IL-5 reverse; 5’- TCT CCG TCT TTC TTC TCC ACA C -3’ (Gene-Bank accession number: NM_000879.3), IL-13 forward; 5’- TGG TAT GGA GCA TCA ACC TGA C -3’, IL-13 reverse; 5’- GCA TCC TCT GGG TCT TCT CG -3’ (Gene-Bank accession number: NM_001354993.2), GAPDH forward; 5’-ATC ACC ATC TTC CAG GAG CGA-3’, GAPDH reverse; 5’-AGG GGC CAT CCA CAG TCT T-3’ (Gene-Bank accession number: NM_001357943.2)
f) It is not clear if the authors also analyzed pure standards in UPLC. If so, could they show a profile with only the standard molecules?
We really agreed with your suggestion. The content analysis of three components need to further application and precise comprehension of L.obtusiloba's biological activity. Regretfully, we did not conduct the content analysis of major components of L.obtusiloba. However, we plan the analysis of contents of three component in L. obtusiloba and, if possible, we should conduct and add the analysis results as supplement data prior to publication or further study using three components. This point also has been described in the limitation of study of Discussion section as follow:
Page 16, L509-515.
Collectively, these results demonstrated the protective effects of LOL on allergic asthma. However, there is some limitations in this study. Firstly, the effects of LOL did not compare with positive control in vitro experiments. For precise comprehension of L. obtusiloba’s biological activities, further investigation applies to positive control in vitro. Secondly, it is yet not known which components possess the anti-inflammatory/anti-oxidant activities in the OVA-challenged asthma model. Hence, it is required to clarify the contents and activities of the three major active components for further therapeutic application on allergic asthma.
g) ROS and RNS should be also measured in this work, to better confirm their data and the hypothesis of redox state modulation
It has been added in Figure 6 and Figure 8 of Results section as follow:
Please see the attachment.
Page 11, L378-L386.
3.6. Effects of LOL on Nrf-2 Pathway, production of ROS and NO, and Oxidative Stress Markers in the Lung Tissue of the OVA-challenged Asthma Model
As presented in Figure 6, OVA-challenged mice showed a significant decrease in nuclear Nrf-2, HO-1, and NQO-1 expression compared to the normal controls. OVA-challenged mice exhibited an increase in levels of ROS, NO, and TBARS, as well as a decrease in activities of GSH, CAT, and SOD compared with normal controls. In contrast, LOL treatment upregulated nuclear translocation of Nrf-2, HO-1, and NQO1 expression compared with the OVA-challenged mice. Moreover, LOL treatment suppressed the productions of ROS, NO, and TBARS, whereas elevated activities of GSH, CAT, and SOD in lung tissues compared with the OVA-challenged mice.
Page 13, L415-Page 14, L424.
3.8. Effects of LOL on p65NF-κB and Nrf-2 Pathway, Oxidative Stress Markers and DPPH Radical Scavenging Activity in TNF-α-stimulated NCI-H292 Cells
As shown in Figure 8, TNF-α-stimulated cells increased the phosphorylation of p65NF-κB and decreased the nuclear translocation of Nrf-2 and HO-1. In addition, TNF-α treatment increased the ROS production and decreased the GSH content and SOD activity in NCI-H292 cells. In contrast, LOL-treated cells inhibited the phosphorylation of p65NF-κB compared with the TNF-α-stimulated cells. LOL treatment increased the translocation of Nrf-2 into the nucleus with elevated HO-1 expression. LOL markedly suppressed ROS production, and increased GSH contents, as well as SOD activities. In addition, LOL exhibited a significant increase in DPPH radical scavenging activity in a concentration-dependent manner.
h) a graphical representation which summarizes the main data obtained by the present work should be added, to collect all together the effects of the plant extract in mice
It has been added in Page 22 and please see the attachment.

Round 2
Reviewer 1 Report
- The positive control must still be provided.
- The contents of three components must still be provided.
Author Response
Reviewer 1
1. The positive control must still be provided.
The results of positive control in vitro have been added in the Figure 7 and Figure 8 with Materials and Methods as follow:
Page 6, L261-266
2.13. Measurement of levels of Pro-inflammatory Cytokine Levels and ROS, and Oxidative Stress Marker in TNF-α-stimulated NCI-H292 Cells
To further therapeutic application of LOL for asthma treatment, TNF-α-stimulated human lung epithelial cells, NCI-H292, has been used to evaluate the anti-inflammatory effects in vitro corresponding to OVA-challenged asthma mice model [42-44]. The cells (5×104 cells/well) were seeded in 6-well plates in RPMI media, treated with LOL (0, 25, 50, and 100 μg/mL) and DEX (20 μg/mL) for 1 h, and then incubated in the presence of human recombinant TNF-α 30 ng/ml for 30 min or 24 h.
Please see the attachment (Figures 7 and 8)
2. The contents of three components must still be provided.
It has been revised in the Results and Table 1 as follow:
Page 7, L301-Page 8, L319
3.1. Tentative Characterization of LOL Extract
LOL were extracted using 100% methanol and analyzed by UPLC Q-TOF/MS (Figure 1 and Table 1). The compounds of LOL were tentatively identified according to information derived from MS deprotonated molecules ([M–H]–) and the fragmentation pattern of mass spectra compared with previous literature in negative mode. In previous studies, the quercetin and kaempferol flavonoids are representative derivatives in many natural products [45-47]. Peak 1 exhibited [M–H]– (447 Da) and [M–146–H]– (301 Da), which were characteristic fragment of quercetin at m/z 301 and rhamnoside (146 Da) in negative mode [46,47]. The fragment patterns of peak 2 demonstrated [M–H]– at m/z 431 and [M–146–H]– at m/z 282. Kaempferol and rhamnoside have molecular weights of 285 and 146, respectively, and peak 2 was tentatively identified as kaempferol rhamnoside [47,48]. The contents of quercetin rhamnoside and kaempferol rhamnoside determined to be 26.04 ± 0.05 mg/g (2.60%) and 15.82 ± 0.02 mg/g (1.58%), respectively (Table 1). Please see the attachment (Table 1).
As you requested, we analyzed the contents of tentatively identified major compounds, including quercetin rhamnoside (quercitrin), kaempferol rhamnoside (afzelin), taxifolin pentoside (taxifolin 3-xyloside) (please see the attachment). However, taxifolin 3-xyloside showed a different RT (5.73 min) compared to previous our result (4.42 min). Thus, we revised Peak 1 (taxifolin pentoside) to “Unknown”. Consistent with previous studies, it has been reported that L. obtusiloba contained two major compounds, quercetin rhamnoside and Kaempferol rhamnoside [48].
48. Hong C.O., Rhee C.H., Won N.H., Choi H.D., Lee K.W. Protective effect of 70% ethanolic extract of Lindera obtusiloba Blume on tert-butyl hydroperoxide-induced oxidative hepatotoxicity in rats. Food Chemical Toxicology. 53:214–220 (2013).
In addition, we attached the standard curves of afzelin and quercitrin in attachment file

Reviewer 2 Report
The authors addressed all requests except that the suggestion reported in the point b. They should add in the introduction the general concepts which were indicated as lacking and also support them with adequate literature, as also suggested.
Author Response
Response to Reviewer’s comments
The authors addressed all requests except that the suggestion reported in the point b.
[b) in the introduction, the authors should report that plant extracts are prone to modulate animal cell oxidative status because it is minimally mentioned. Moreover, the concept of phytocomplex should be reported. I suggest to support such declarations with the following citations: Phytotherapy Research, 2010, 24(4), 595-601; Phytomedecine, Volume 46, 2018, Pages 1-10; Pharmacognosy research, 2016, 8.Suppl 1: S42; European Journal of Pharmaceutical Sciences, 2017, 96, 53-61.]
They should add in the introduction the general concepts which were indicated as lacking and also support them with adequate literature, as also suggested.
It has been described in the Introduction as follow :
Page 2, L81-90
Oxidative stress play a crucial role in pathological conditions, including inflammation, cancer, metabolic and immune disorders [29,30]. Many researcher has been reported that plant extracts are prone to modulate cellular oxidative status in several pathological conditions as mentioned above [30-32]. The bioactive phytochemicals or secondary metabolites from plant extracts possess various biological activities, such as antioxidant, anti-inflammatory, and antitumor both in vitro and in vivo [31,32]. Recently, with the current upsurge of interest in efficacy of plant extracts, active substance extracted and concentrated from plants (as single or combined phytocomplexes), such as polyphenols, flavonoids, and alkaloids, have been extensively investigated to proven their clinical efficacy for pharmaceutical application [29,32]. Further, it can be used to prevent or cure pathological conditions when they have proven safety and beneficial properties, as well as better bioavailability [29].
-
- Santini A., Tenore G.C., Novellino E. Nutraceuticals: a paradigm of proactive medicine. European Journal of Pharmaceutical Sciences. 1(96):53-61 (2017).
- Nanni V., Canuti L., Gismondi A., Canini A. Hydroalcoholic extract of Spartium junceum L. flowers inhibits growth and melanogenesis in B16-F10 cells by inducing senescence. Pytomedicine. 15(46)1-10 (2018).
- Nardi G.M., Januario A.G.F., Freire C.G., Megiolaro F., Schneider K., Perazzoli M.R.A., Nascimento S.R.D., Gon A.C., Mariano L.N.B., Wagner G., Niero R., Locatelli C. Anti-inflammatory activity of Berry fruits in mice model of inflammation is based on oxidative stress modulation. Pharmacoqnosy research. 8(1):S42-49 (2016).
- Gutierrez R., Alvarado J.L., Presno M., Perez-Veyna O., Serrano C.J., Yahuaca P. Oxidative stress modulation by Rosmarinus officinalis in CCI4-induced liver cirrhosis. Phytotherapy research. 24(4):595-601 (2010).

Round 3
Reviewer 1 Report
Authors have answered all comments. This manuscript is acceptable now.